# Characterization of Small Extracellular Vesicles Isolated from *Aurelia aurita*

**DOI:** 10.3390/biology14080922

**Published:** 2025-07-23

**Authors:** Aldona Dobrzycka-Krahel, Aleksandra Steć, Grzegorz S. Czyrski, Andrea Heinz, Szymon Dziomba

**Affiliations:** 1Business Faculty, WSB Merito University in Gdansk, Al. Grunwaldzka 238 A, 80-266 Gdansk, Poland; aldona.dobrzycka-krahel@gdansk.merito.pl; 2Department of Toxicology, Faculty of Pharmacy, Medical University of Gdansk, 107 Hallera Street, 80-416 Gdansk, Poland; aleksandra.stec@gumed.edu.pl; 3LEO Foundation Center for Cutaneous Drug Delivery, Department of Pharmacy, University of Copenhagen, 2100 Copenhagen, Denmark; grzegorz.czyrski@sund.ku.dk (G.S.C.); andrea.heinz@sund.ku.dk (A.H.)

**Keywords:** capillary electrophoresis, extracellular vesicles, moon jellyfish, marine organism, size-exclusion chromatography

## Abstract

Extracellular vesicles (EVs) are spherical nanoparticles that are secreted by living cells. These structures are currently attracting significant attention due to their unique biological properties. However, the properties of EVs secreted by jellyfish remain unexplored. For the first time, we demonstrate a preliminary physicochemical characterization of EVs derived from moon jellyfish oral arm samples. We also present an efficient isolation method for the purification of EVs from jellyfish tissue. Moreover, we show that these structures carry nucleic acids and are, most likely, highly glycated. All these observations indicate the potential biological activity of moon jellyfish-derived EVs.

## 1. Introduction

Jellyfish, belonging to the phylum Cnidaria, are marine organisms known for their unique physiology and relatively simple body structure [1,2]. These organisms possess the ability to rapidly regenerate damaged tissues [3,4] and even reconstruct organs de novo under special culture conditions [5]. “The immortal jellyfish” *Turritopsis dohrnii* (Weismann, 1883) has the ability to reverse age, with its medusa being able to metamorphose back into the polyp stage in the phenomenon of “reverse development”. This species can regenerate its cells and could theoretically escape death [6]. Therefore, jellyfish have attracted the attention of a large scientific community working on anti-aging and regenerative biomaterials that could potentially have applications in human medicine. To date, research has mainly focused on collagen-based biomedical applications of jellyfish [7,8].

Jellyfish also represent a source of other bioactive compounds with diverse biomedical applications. Glycosaminoglycans of jellyfish have anticoagulant, anti-inflammatory, and immunomodulatory properties [8]. Green fluorescent protein obtained from jellyfish has revolutionized molecular biology by enabling the real-time tracking of cellular processes, and it is widely used in diagnostics [9]. Some jellyfish species produce venom containing peptides and proteins with potential in developing painkillers, anticancer agents, and antimicrobial drugs [10].

Recent advancements in medicine have highlighted the crucial role of extracellular vesicles (EVs) in cellular communication, tissue repair, and regeneration [11,12]. EVs are small, membrane-bound vesicles secreted by cells that carry bioactive molecules, e.g., proteins and nucleic acids, and play a key role in modulating inflammation, promoting cell proliferation, and enhancing tissue remodeling during wound healing processes [13]. While research on EVs has predominantly focused on mammalian sources, such as mesenchymal stem cells [14], there is growing interest in exploring alternative, non-mammalian, aquatic invertebrate EV sources with unique regenerative properties [15]. In aquatic invertebrates, EVs are present and may act similarly to those in mammalian vertebrates, but research in this area is limited and still unexplored [15].

The exploration of jellyfish-derived EVs represents a novel and promising research area that may contribute to advanced regenerative therapies. The jellyfish tissue structure resembles the architecture of human skin tissue, which makes jellyfish a relevant object in scientific research [16]. The object of our study, the moon jellyfish *Aurelia aurita* (Linnaeus, 1758), exhibits remarkable regenerative abilities [17,18], suggesting that its body components may promote tissue repair. The first reports have emerged confirming the regenerative potential of EVs isolated from other Cnidaria species [19,20]. Due to the gap in knowledge regarding the EVs of Cnidarians, the aim of our study was to develop a methodology for EV isolation from *A.aurita* oral arm samples, characterize the isolated vesicles, and determine the limitations related to the biomedical application of these structures.

## 2. Materials and Methods

### 2.1. Chemicals

Disodium tetraborate decahydrate, bovine serum albumin (BSA), phosphate-buffered saline (PBS), and sodium dodecyl sulfate (SDS) were obtained from Merck (Steinheim, Germany). Sodium hydroxide was purchased from Avantor (Gliwice, Poland). All chemicals were of analytical grade. In all the experiments, deionized water was used.

### 2.2. Sample Collection

The material was collected from the nearshore zone of the Gulf of Gdansk (Baltic Sea, Poland) in the yacht port in Gdynia, on 22 July 2024, using a hand net. The oral arms of an adult medusa of *A. aurita* were cut off with a scalpel, and 0.5 mL of each sample was immediately placed in an Eppendorf tube. The samples were stored in a freezer at −20 °C until further laboratory analysis.

### 2.3. Isolation of EVs

Samples were thawed at room temperature and centrifuged for 30 min at low speed (4000 RCF). The supernatant was collected and centrifuged at 25,000 RCF. All centrifugation steps were performed in a Sorvall 16R centrifuge (Thermo Fisher Scientific, Waltham, MA, USA) using a fixed-angle rotor. An amount of 0.5 mL of supernatant was fractionated using 35 nm qEVoriginal Gen 2 size-exclusion chromatography (SEC) columns (IZON, Christchurch, New Zealand) according to the vendor’s recommendations. PBS solution was used for column equilibration and sample elution. Nine fractions (0.5 mL each) were collected. Fractions 7, 8, and 9 were combined and concentrated with an Amicon Ultra 0.5 centrifugal filter (Merck) to a final volume of about 0.2 mL. The isolates were stored at 4 °C until further use.

### 2.4. Bicinchoninic Acid Assay (BCA)

The protein content in isolates was determined using a Pierce BCA kit (Thermo Fisher Scientific). The samples and standards were mixed with 6% (*w*/*w*) SDS solution in a 9:1 volumetric ratio, and 10 µL of this mixture was transferred to a 96-well plate. An amount of 200 µL of BCA reagent mixture was added to each well, and the plates were incubated at 37 °C for 30 min to accelerate the reaction. The absorbance in each well was measured at 562 nm using an Infinite 200 plate reader (Tecan, Männedorf, Switzerland). The protein content was determined based on the calibration curve constructed with the BSA solution standard (100–2000 µg mL^−1^).

### 2.5. Tunable Resistive Pulse Sensing (TRPS)

TRPS experiments were performed with an Exoid (IZON) using an NP80 nanopore (IZON) device and the CPC100 calibration standard (IZON). A three-point calibration method was used. The stretch was adjusted to 47 mm, and a voltage of 600 mV was used to achieve the desired sensitivity with the method. The number-weighted distribution of particle diameters in each measurement was averaged and plotted in histogram format with a bin size of 5 nm.

### 2.6. Electrophoretic Light Scattering (ELS)

The measurement of zeta potential was performed with the Zetasizer Nano ZS (Malvern Instruments, Malvern, Worcestershire, UK). U-curved disposable cuvettes (DTS1070; Malvern Instruments) were used. The refractive index of the material and water was set to 1.45 and 1.33, respectively. The viscosity of the dispersant was 0.8872 cP (mPa s). The material absorption was 0.001. EV isolates were diluted 20-fold with deionized water for measurement. The zeta potential was calculated with the Hückel model. At least four measurements were taken for each sample.

### 2.7. Capillary Electrophoresis (CE)

CE experiments were performed using a PACE MDQ plus system (Sciex, Framingham, MA, USA) equipped with a laser-induced fluorescence (LIF) detector. Electrophoresis was conducted in fused silica capillaries (50 µm i.d. × 363 µm o.d. × 30.2 cm of total length) using a positive voltage of 10 kV. The background electrolyte (BGE) was composed of 25 mM borate buffer (pH 9.2) and was filtered through a 0.22 µm PES syringe filter (Whatman, Little Chalfont, UK). During the analysis, the temperature was kept constant (25 °C).

At the beginning of each working day, the capillary was sequentially rinsed with a 0.1 M solution of NaOH, water, and BGE (30 min each). Before every run, the capillary was conditioned with a 0.1 M solution of NaOH (1 min), water (1 min), and BGE (2 min). All rinsing steps were performed at a pressure of 137.9 kPa. The sample injection was preceded by a water dipping procedure. A short post-injection plug of BGE was applied (5 s, 3.45 kPa).

Samples were stained with SYBR Gold dye (Thermo Fisher Scientific) [21,22]. A stock solution of SYBR Gold was diluted 100-fold with dimethyl sulfoxide, mixed with the sample in a 1:19 volumetric ratio, and incubated at room temperature in the dark for 1 h. Samples were injected into the CE system hydrodynamically (5 s, 3.45 kPa). During CE analysis, detection was performed using a 488 nm laser and a 520 nm emission filter.

The zeta potential of EVs was determined with ELS and CE measurements. In CE, the zeta potential of EVs was calculated using Equations (S1)–(S3) (Appendix A).

### 2.8. Cryogenic Transmission Electron Microscopy (Cryo-TEM)

The morphology of the isolated EVs was examined using cryo-TEM. To achieve this, 3 µL of the isolate was applied to a discharged Lacey formvar/silicon monoxide 300 mesh copper grid (Ted Pella Inc., Redding, CA, USA). The grid was then vitrified via immersion in liquid ethane with a Vitrobot Mark IV (Thermo Fisher, MA, USA). Images of the EVs were taken at various magnifications using an FEI Tecnai G2 20 TWIN transmission electron microscope, operated at 200 keV in a low-dose mode, and an FEI High-Sensitivity Eagle camera.

## 3. Results

EVs were isolated from *A. aurita* oral arm samples according to the procedure described in Section 2.3. The protein content in the isolates, measured with BCA, was below the lower limit of quantification (<0.1 mg mL^−1^). However, the total protein content in *A. aurita* oral arm samples was relatively low (0.43 ± 0.02 mg mL^−1^), which justifies the even lower protein concentration in the EV isolates.

TRPS analysis confirmed the presence of particles in the small EV-characteristic size distribution range (70–150 nm; Figure 1A). The mean, mode, and median size (±SD) of the vesicles were equal to 97 ± 5 nm, 84 ± 9 nm, and 90 ± 6 nm, respectively (three measurements of two independently obtained isolates).

Cryo-TEM analysis confirmed the presence of spherical particles (Figure 1B) within the TRPS-determined size distribution (Figure 1A). The images present round-shaped structures with a clearly distinguishable membrane. In some images, multilayer-like vesicles are shown, which might be due to the EVs overlapping during the analysis [23].

Samples stained with SYBR Gold dye and CE-LIF analysis showed the presence of a relatively broad signal (<20,000 theoretical plates m^−1^) characteristic of EVs (Figure 1C) [22,24]. The effective staining of EVs with SYBR Gold confirmed that these structures carry nucleic acids. The average electrophoretic mobility of the EVs determined with CE measurements was −3.30 ± 0.02 × 10^−8^ m^2^ s^−1^ V^−1^ (two measurements of two independently obtained isolates). The measured electrophoretic mobility was used for the calculation of zeta potential (Equations (S1)–(S3)), which was equal to −71.6 ± 0.6 mV.

The zeta potential of the isolated EVs was measured with the ELS technique (Figure 1D). The measured average values ranged from −17.3 to −21.6 mV with a mean of −19.8 ± 1.6 mV (four measurements of two independently obtained isolates). The average electrophoretic mobility of EV determined with ELS was −1.03 ± 0.08 × 10^−8^ m^2^ s^−1^ V^−1^.

## 4. Discussion

Jellyfish are generally widely distributed and occur in large biomasses, playing an important ecological role as an energy source both in pelagic and deep-sea food webs [25]. To date, research on *A. aurita*, commonly known as the moon jellyfish, has primarily focused on its life cycle [26,27], population dynamics [28,29], interactions with other marine species [30], and responses to environmental changes [31,32]. The species has also been studied for its bioactive compounds with potential applications in medicine, biotechnology, and cosmetics, and it has been confirmed that it is a source of polysaccharide compounds, collagen, and amino acids [8,33,34]. 

Since the research on EVs of aquatic animals is a relatively novel and poorly explored concept [15], our study has focused on the development of an EV isolation methodology, the preliminary characterization of vesicles, and the determination of the major limitations of the research. In the study, we implemented differential centrifugation, SEC, and ultrafiltration (UF) for the isolation of EVs. Similar methodology has already been shown to be effective in the isolation of vesicles from plant material [21], cell cultures [35], human plasma samples [36], and other sources. The limited sample volume excluded initial sample preconcentration for yield improvement. As a result, the EV content in the finally obtained isolates was relatively low. For instance, the total protein content in individual isolates of moon jellyfish *A. aurita* was below the lower limit of quantification of the BCA test. This observation is justified by the relatively low protein level in raw samples (0.43 ± 0.02 mg mL^−1^). Nonetheless, the low protein yield achieved did not allow for a reliable proteomics analysis of the isolates. The implementation of mass spectrometric analysis might shed light on the biological role played by the EVs in jellyfish. Within the Cnidaria phylum, extensive proteomics analysis is only reported for EVs isolated from *Hydra vulgaris* [19]. The study revealed 52 vesicular proteins related to EV biogenesis, structural components (e.g., cytoskeleton, microfibril), transcription, metabolic control, adhesion, and signaling.

The size distribution of EVs varies depending on the source, isolation methodology, and analytical technique used for measurement. In our study, EVs were within the 70 –150 nm size range (Figure 1A). It should be emphasized that the measurement of the smallest particles was limited by the sensitivity of the TRPS instrument. For comparison, the application of cryo-TEM enabled the visualization of particles below 50 nm in diameter. A similar particle size distribution (50–150 nm) was determined with microfluidic resistive pulse sensing for EV isolates obtained from *H. vulgaris* [19].

Attention should also be paid to the fact that this study was focused on small EVs (exosome-like vesicles). Although identity confirmation of the EVs by proteomics analysis was not possible, a relatively narrow size distribution in TRPS measurements and in cryo-TEM images (similar EVs were demonstrated with TEM by Moros and coworkers for EVs obtained from *H. vulgaris* [19]) confirms the homogeneity of the isolates.

The transfer of nucleic acids by EVs obtained from *A. aurita* suggests that the vesicles can suppress or modulate the production of specific proteins in target cells, priming the immune system to mount defenses against diseases and deliver therapeutic payloads efficiently [37]. Shinzato et al. demonstrated that genes related to exosome secretion were up-regulated in *Acropora digitifera* when the polyp was exposed to bacterial infection under thermal stress, which may confirm the role of EVs in immune response in Cnidaria [38]. Transcriptomic analysis of EVs released by *H. vulgaris* revealed genes modulating developmental processes [19]. The authors demonstrated EV internalization between individuals and a modulatory effect of EVs on head and foot regeneration in Hydra. Regenerative properties of EVs isolated from sea anemone *Aulactinia stella* tissue have also been reported [20].

A tremendous difference between the zeta potential values of EVs measured with ELS and CE (−19.8 ± 1.6 mV and −71.6 ± 0.6 mV, respectively) was observed. It should be emphasized that the calculation of zeta potential with CE using Equations (S1)–(S3) is not accurate as it does not include relaxation and retardation effects observed during the electrophoresis of highly charged particles [24,39]. Nevertheless, this simple estimation is sufficient to demonstrate the discrepancy between ELS and CE experiments. The results may be attributed to the experimental conditions. The CE measurement of SYBR Gold-stained vesicles was performed in a background electrolyte, while ELS analysis was conducted in a low-ionic-strength medium (20-fold-diluted PBS solution) in a native state. However, a significantly higher ionic strength of the medium in CE was expected to decrease the zeta potential [40], and staining with SYBR Gold was shown not to affect the physicochemical properties of EVs [22]. The explanation behind these phenomena might be related to the ability of borate ions (used as a background electrolyte component in CE) to complex cis-diol moieties that are typically abundant in glycans. Assuming that this hypothesis is true, the observation implies that a highly glycated form of small EVs is secreted by *A. aurita*. Various glycation degrees of EVs, as well as their impact on the zeta potential and electrophoretic mobility of vesicles, have already been demonstrated [41]. The role of EV glycation in Cnidaria has not been investigated yet. It may be a consequence of a relatively high content of polysaccharides in these organisms [7,42]. Attention should be paid to the fact that in biological environments, such artificial complexation would not occur. The ELS measurements were performed in a more physiologically relevant buffer and likely reflect a more accurate representation of the EVs’ native surface charge. Thus, the greater zeta potential observed with borate buffer remains irrelevant to EV stability in vivo. Besides the biological significance of this observation, the question regarding the medical utility of this observation remains open. Jellyfish are considered an alternative source of collagen [43], showing antioxidant, anti-inflammatory, and immunomodulatory properties [44,45]. An assessment of these properties of jellyfish-derived EVs might be interesting, especially considering that the active components are a part of a nanoscale carrier (EVs), which modulates transport through biological barriers and can increase the bioavailability of these compounds. Nevertheless, the presented preliminary study shows that biological experiments on EVs isolated from *A. aurita* are possible only when the isolation process is scaled up.

## 5. Conclusions

The method demonstrated here was found to be efficient in the isolation of small EVs from *A. aurita* oral arm samples. The experiments and the literature indicate that these structures are nanoscale containers of biologically active components. A better understanding of their biological role and an assessment of their beneficial activity will only be possible after an improvement in isolation yield. To achieve this goal, the process will have to be scaled up.

## Figures and Tables

**Figure 1 biology-14-00922-f001:**
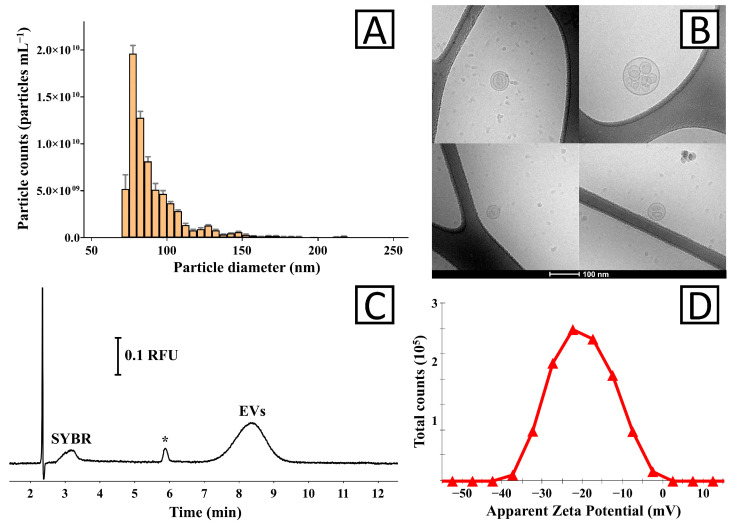
Characterization of EV isolates obtained from *Aurelia aurita* oral arm samples. (**A**) The size distribution of isolated vesicles was determined using TRPS. (**B**) Cryo-TEM image showing the morphology of EVs. The bar corresponds to 100 nm. Raw images and a zoomed-out image are provided in Appendix A. (**C**) CE-LIF analysis of the isolate stained with SYBR Gold dye. The residue of free dye is marked in the figure (SYBR). The asterisk (*) indicates an unidentified impurity. (**D**) Zeta potential of the EV isolate determined with the ELS technique.

## Data Availability

The original contributions presented in this study are included in the article/Appendix A. Further inquiries can be directed to the corresponding author.

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
