# Peer review of "Characterization of Small Extracellular Vesicles Isolated from Aurelia aurita"

_biology, 2025, doi:10.3390/biology14080922_

Round 1
Reviewer 1 Report
Comments and Suggestions for Authors
The study explores an understudied EVs source from Aurelia aurita, which is new and potentially relevant to regenerative medicine. A well-defined multi-step isolation procedure comprising differential centrifugation, SEC, and ultrafiltration is employed, reflecting technical competence. However I do have some concerns here:
- In the research, the yield of EVs from A. aurita was so low that omics or biological activity analysis was not possible. This limits the depth of functional conclusions.
- While the presence of nucleic acids is confirmed, no data regarding protein composition or EV markers (e.g., CD9, CD63) are provided. This undermines confirmatory evidence of EV identity and biological relevance. they do not yet meet the full criteria for definitive classification as exosomes or sEVs per ISEV/MISEV standard.
- High variation between ELS and CEalignated zeta potential is reported but not fully explained, reducing the extent of understanding of EV surface properties.
- The paper explores the jellyfish EVs' regenerative and immunomodulatory capacities but also avoids even preliminary bioassays.
- There are limited biological replicates and total sample number. The sample volume is merely 0.5 mL, which raises reproducibility and robustness issues.
- Figure 1, while informative, can be improved with clearer labels (e.g., particle counts, supplementary figures' scale bars).
- Even rudimentary in vitro bioactivity assays (e.g., fibroblast proliferation, wound healing scratch assay) would be worth-adding and substantiate regenerative claims.
- Minor matters of grammatical error and a few clumsy phrasings ("minute protein concentration") can be polished for fluency and comprehensibility.
- A reference to observed trends in regenerative assays (if done) would add biomedical significance outlined in the intro.
Author Response
1. The study explores an understudied EVs source from Aurelia aurita, which is new and potentially relevant to regenerative medicine. A well-defined multi-step isolation procedure comprising differential centrifugation, SEC, and ultrafiltration is employed, reflecting technical competence. However I do have some concerns here:
In the research, the yield of EVs from A. aurita was so low that omics or biological activity analysis was not possible. This limits the depth of functional conclusions.
We agree with the reviewer. Due to the limited availability of the specimens, we were not able to possess enough material for the experiments to perform more complex analyses. We collected only three samples (non-lethal sample collection) and obtained three isolates (about 100 μL each). Moreover, the concentration of EVs in the isolates was too low to perform omics analysis. We are aware of the limitations of the conducted study, which are emphasized in the manuscript (e.g., in the Conclusion). While the EV biology of marine organisms (especially jellyfish) is a relatively new and unexplored area, we decided to publish the results we obtained in the form of a short communication. We hope that this preliminary study will be found useful for other researchers who can, for instance, implement the methodology developed by our group.
2. While the presence of nucleic acids is confirmed, no data regarding protein composition or EV markers (e.g., CD9, CD63) are provided. This undermines confirmatory evidence of EV identity and biological relevance. they do not yet meet the full criteria for definitive classification as exosomes or sEVs per ISEV/MISEV standard.
We agree with the reviewer that, based on the presented results, it is not possible to classify the isolated vesicles as exosomes or, e.g., small ectosomes. According to ISEV recommendations, the detection of certain proteomic markers is required for the classification of EVs. We assume that it would be very interesting to perform a complete untargeted MS analysis of the isolates to have a deep insight into the proteomic composition of jellyfish-derived EVs. According to our knowledge, this would be the first analysis of this kind of sample. Unfortunately, due to the limited availability of material used in the study, we were not able to perform this type of analysis. Hopefully, we will be able to extend our research in the very near future.
3. High variation between ELS and CE alignated zeta potential is reported but not fully explained, reducing the extent of understanding of EV surface properties.
The difference between zeta potential values is high. It is not clear what the reason is for so high variation. We hypothesize that this observation might be due to the presence of polysaccharides on the EV surface and their complexation with the BGE component (disodium tetraborate). It has already been demonstrated that the e.g., sialylation of EV surface or presence of phosphatidylserine has a significant impact on zeta potential/electrophoretic mobility of EVs [Japanese Journal of Applied Physics 53, (2014) 06JL01; Journal of Extracellular Vesicles 8 (2019) 1579541]. Complete explanation of this phenomenon in jellyfish-derived EVs will require additional experiments with the use of advanced omics analyses. While this type of investigation was not possible in our study because of the limited availability of biological material, we disclose all information we obtained in the form of a short communication. Whole discussion on zeta potential measurements can be found in lines 230 - 261:
“A tremendous difference between the zeta potential values of EVs measured with ELS and CE (-19.8 ± 1.6 mV and -71.6 ± 0.6 mV, respectively) was observed. It should be emphasized that the calculation of zeta potential with CE using Equations 1 – 3 is not accurate as it does not include relaxation and retardation effects observed during electrophoresis of highly charged particles 24,39. Nevertheless, this simple estimation is sufficient to demonstrate the discrepancy between ELS and CE experiments. The results may be attributed to the experimental conditions. The CE measurement of SYBR Gold-stained vesicles was performed in a background electrolyte, while ELS analysis was conducted in a low ionic strength medium (20-fold diluted PBS solution) in a native state. However, a significantly higher ionic strength of the medium in CE was expected to decrease the zeta potential 40, and staining with SYBR Gold was shown not to affect the physicochemical properties of EVs 22. The explanation of these phenomena might be related to the ability of borate ions (used as background electrolyte component in CE) to complex cis-diol moieties that are typically abundant in glycans. Assuming that this hypothesis is true, the observation implies a highly glycated form of small EVs secreted by A. aurita. Various glycation degrees of EVs, as well as their impact on zeta potential and electrophoretic mobility of vesicles, have already been demonstrated 41. The role of EV glycation in Cnidaria has not been investigated yet. It may be a consequence of a relatively high content of polysaccharides in these organisms 7,42. The attention should be paid to the fact that in biological environments, such artificial complexation would not occur. The ELS measurements were performed in a more physiologically relevant buffer and likely reflect a more accurate representation of the EVs' native surface charge. Thus, the greater zeta potential observed with borate buffer remains irrelevant for EV stability in vivo. Besides the biological significance of this observation, the question regarding the medical utility of this observation remains open. Jellyfish are considered an alternative source of collagen 43, showing antioxidant, anti-inflammatory, and immunomodulatory properties 44,45. The assessment of these properties of jellyfish-derived EVs might be interesting, especially considering that the active components are a part of a nanoscale carrier (EVs), which modulates the transport through biological barriers and can increase bioavailability of these compounds. Nevertheless, the presented preliminary study shows that the biological experiments on EVs isolated from A. aurita can be possible only when the isolation process is scaled up.”
4. The paper explores the jellyfish EVs' regenerative and immunomodulatory capacities but also avoids even preliminary bioassays.
Unfortunately, we cannot agree with the reviewer. In the manuscript, we discuss the unique abilities of jellyfish to regenerate their tissues. We also refer to a few studies on the properties of EVs isolated from cnidarian species. We hypothesize that the EVs derived from the moon jellyfish might have similar properties. However, biological activity evaluation was out of the scope of this short communication due to technical limitations.
5. There are limited biological replicates and total sample number. The sample volume is merely 0.5 mL, which raises reproducibility and robustness issues.
The aim of this pilot study was to develop a method for the isolation of EVs from moon jellyfish and to provide preliminary physicochemical characterization of jellyfish-derived EVs. The samples were collected non-lethally. Small sample volume resulted from the size of the animal. Collection of a greater amount of the tissue might significantly impair the organism and decrease the chance for survival before its complete regeneration. Furthermore, we wanted to reduce the number of animals exploited in our study. We follow the 3Rs (Replacement, Reduction and Refinement) approach which is required for experiments on live animals by law in EU countries [eBioMedicine 76 (2022) 103900 – The 3Rs of Animal Research; Frontiers in Veterinary Science 10 (2023) – Advancing the 3Rs: innovation, implementation, ethics and society]. The 3Rs rule includes:
- Replacement - avoiding or replacing the use of animals in areas where they otherwise would have been used.
- Reduction - minimizing the number of animals used, consistent with scientific aims.
- Refinement - minimizing the pain, suffering, distress, or lasting harm that research animals might experience.
We are aware of the fact that the number of samples we used in the study is not representative enough to, e.g., perform the validation of the method. However, in our opinion, it was sufficient to demonstrate the proof of concept.
6. Figure 1, while informative, can be improved with clearer labels (e.g., particle counts, supplementary figures' scale bars).
We appreciate the reviewer’s helpful comment. Figure 1 has been revised to include clearer labels in line with the suggestion. Regarding the supplementary figures, we would like to note that the scale bars used are relatively large, well visible, and represent the sizes depicted in the zoom-out images. If there are specific concerns about these figures that we may have overlooked, we would be grateful if the reviewer could clarify them so that we can make the necessary adjustments.
7. Even rudimentary in vitro bioactivity assays (e.g., fibroblast proliferation, wound healing scratch assay) would be worth-adding and substantiate regenerative claims.
We fully agree with the reviewer that in vitro bioactivity assays, such as fibroblast proliferation or wound healing scratch assays, would significantly strengthen the study and help substantiate any regenerative claims. However, due to the very limited quantity of EVs obtained in this study, it was not feasible to perform such assays. Our priority was to use the available material for detailed characterization, which we believe is an essential first step, given the scarcity of studies on EVs from Cnidarians. We consider bioactivity assays a critical direction for future research and are currently working on optimizing our isolation protocol to enable sufficient yield for functional studies.
8. Minor matters of grammatical error and a few clumsy phrasings ("minute protein concentration") can be polished for fluency and comprehensibility.
The whole manuscript was proofread, and the minor grammatical corrections were made. The corrections made in the manuscript were highlighted.
9. A reference to observed trends in regenerative assays (if done) would add biomedical significance outlined in the intro.
We appreciate the reviewer’s comment and agree that including data from regenerative assays would indeed strengthen the biomedical significance of our findings. However, as noted earlier, we were unable to perform such assays due to the very limited quantity of EVs obtained. Our primary goal in this initial study was to isolate and characterize EVs from Aurelia aurita, which remains a rarely studied source of extracellular vesicles. We believe that this foundational work is essential to establish a platform for future studies.
Recognizing the importance of regenerative applications, we plan to investigate the bioactivity and therapeutic potential of moon jellyfish-derived EVs in follow-up studies, once we are able to improve EV yield and scale up our experiments.
Reviewer 2 Report
Comments and Suggestions for Authors
This study presents a novel methodology for isolating and characterizing small extracellular vesicles (EVs) from the moon jellyfish Aurelia aurita, providing insights into their physical properties and potential biomedical applications. Below, I outline a series of observations and questions for the authors to address in an organized manner:
1-The protein yield from the isolated EVs is very low, have the authors considered alternative isolation techniques to improve yield for future omics analyses?
2- Can the authors discuss what are some implications of this disparity on the stability of the EVs in biological environments?
3-Have the authors conducted any biological assays (cellular uptake, cytotoxicity, or bioactivity) with these isolated EVs even at low yield?
Comments on the Quality of English LanguageThe manuscript is overall well-written and clearly presented. However, minor proofreading is recommended to improve grammatical accuracy and readability. For example:
The sentence “Additionally, the difference in zeta potential values measured with ELS and CE indicate high glycation of the analyzed vesicles.” should be revised to:
“Additionally, the difference in zeta potential values measured with ELS and CE indicates high glycation of the analyzed vesicles.”
Similarly, the sentence “The species is able to regrow its cells, and could theoretically escape death.” can be improved to: “This species can regenerate its cells and could theoretically escape death.” the enlargement of the isolation scale: scaling up the isolation process..
Author Response
This study presents a novel methodology for isolating and characterizing small extracellular vesicles (EVs) from the moon jellyfish Aurelia aurita, providing insights into their physical properties and potential biomedical applications. Below, I outline a series of observations and questions for the authors to address in an organized manner:
1. The protein yield from the isolated EVs is very low, have the authors considered alternative isolation techniques to improve yield for future omics analyses?
We appreciate this valuable comment. In our opinion, the main problem of the low yield results from the limited amount of biological material. In our research, we have collected three samples. The collection was non-lethal for the jellyfish. Thus, the tissue collected was only a small part of the whole organism. For comparison, in the work of Moros et al. 250 polyps were used for the single isolation (Front. Cell Dev. Biol. 2021, 9, 788117). Also, the protein measurement level conducted in our study showed that the protein level in jellyfish tissue was relatively low. Thus, we assume that the main problem refers to the relatively low content of EVs in the biological material. On the other hand, carrying out the study using a larger amount of biological material would certainly require modification of the isolation method, e.g. in order to preconcentrate the starting material. For this purpose, ultracentrifugation or ultrafiltration techniques might be found suitable, but the application of SEC would still be recommended. It is because SEC (IZON 35 nm columns) efficiently separates vesicles from proteins and low molecular-weight impurities. In our research, it was shown that the EV-rich SEC fractions were devoid of any additional signals indicating co-isolation of sample matrix components. Nevertheless, the modification of the methodology will require further optimization of the isolation parameters.
2. Can the authors discuss what are some implications of this disparity on the stability of the EVs in biological environments?
We thank the reviewer for this insightful comment. We observed that zeta potential values measured by CE were more than twice as high as those obtained using ELS. This disparity is likely attributable to differences in the measurement environments: specifically, the use of borate buffer in CE, which may form complexes with glycoconjugates on the EV surface, thereby increasing the net negative surface charge. This effect suggests a high degree of glycosylation on the vesicle surface.
In biological environments, however, such artificial complexation would not occur. The ELS measurements were performed in a more physiologically relevant buffer and likely reflect a more accurate representation of the EVs' native surface charge. Since zeta potential is a key indicator of colloidal stability, these findings imply that while the EVs may appear highly stable in certain experimental conditions (e.g., in borate buffer), their actual stability in biological fluids may be lower. This highlights the importance of assessing surface charge in physiologically relevant media when predicting EV behavior in vivo, including aggregation potential, circulation time, and cellular uptake efficiency.
We added a short discussion of these implications in the revised manuscript (lines 249 – 252):
“The ELS measurements were performed in a more physiologically relevant buffer and likely reflect a more accurate representation of the EVs' native surface charge. Thus, the greater zeta potential observed with borate buffer remains irrelevant for EV stability in vivo.”
3. Have the authors conducted any biological assays (cellular uptake, cytotoxicity, or bioactivity) with these isolated EVs even at low yield?
We appreciate the reviewer’s insightful question. Unfortunately, due to the extremely low yield of EVs and volume of the isolates (approximately 100 µL), we were unable to perform biological assays such as cellular uptake, cytotoxicity, or bioactivity testing. The entire sample volume was required for thorough physicochemical characterization. Given the novelty of EV isolation from Cnidarians and the technical challenges associated with obtaining sufficient material, we prioritized characterization to establish a foundational understanding. Nonetheless, we recognize the importance of functional studies and plan to address this in future work. Publication of our preliminary results will help us to obtain funding to continue the research and to optimize the isolation protocol to increase EV yield.
4. The manuscript is overall well-written and clearly presented. However, minor proofreading is recommended to improve grammatical accuracy and readability. For example:
The sentence “Additionally, the difference in zeta potential values measured with ELS and CE indicate high glycation of the analyzed vesicles.” should be revised to:
“Additionally, the difference in zeta potential values measured with ELS and CE indicates high glycation of the analyzed vesicles.”
Similarly, the sentence “The species is able to regrow its cells, and could theoretically escape death.” can be improved to: “This species can regenerate its cells and could theoretically escape death.” the enlargement of the isolation scale: scaling up the isolation process..
The whole manuscript was proofread, and the minor grammatical corrections were made. All reviewers’ suggestions were accepted and implemented. The corrections made in the manuscript were highlighted.
Round 2
Reviewer 1 Report
Comments and Suggestions for Authors
Thank you for answering my questions. good luck